# Bioaccumulation of Lanthanum by Two Strains of Marine Diatoms *Nanofrustulum shiloi* and *Halamphora kolbei*

**DOI:** 10.3390/biology14111489

**Published:** 2025-10-24

**Authors:** Daria Sergeevna Balycheva, Anastasiia Andreevna Blaginina, Vyacheslav Nikolaevich Lishaev, Sergey Victorovich Kapranov, Ekaterina Sergeevna Miroshnichenko, Svetlana Nikolaevna Zheleznova, Mikhail Vitalievich Simokon, Vitaliy Ivanovich Ryabushko

**Affiliations:** 1A.O. Kovalevsky Institute of Biology of the Southern Seas of RAS, 299011 Sevastopol, Russia; aablagini@ibss-ras.ru (A.A.B.); megarafik@gmail.com (V.N.L.); sergey.v.kapranov@yandex.ru (S.V.K.); inerlim@gmail.com (E.S.M.); zheleznovasveta@yandex.ru (S.N.Z.); rabushko2006@yandex.ru (V.I.R.); 2Pacific Fisheries Research Center (TINRO-Center), 4 Shevchenko Lane, 690091 Vladivostok, Russia; mikhail.simokon@tinro.vniro.ru

**Keywords:** Bacillariophyta, *Nanofrustulum shiloi*, *Halamphora kolbei*, rare earth elements, bioabsorption, bioadsorption, EDX-spectroscopy, ICP-MS

## Abstract

**Simple Summary:**

Currently, the mining and use of rare earth elements are steadily increasing. Diatoms are known to absorb various elements from the aquatic environment by adsorbing them onto the surface of their frustules and accumulating them within the cell. While there is a considerable amount of data on the bioaccumulation of various metals by diatoms, there is almost no information on the concentration of rare earth elements in their cells. Thus, the aim of this study was to determine the ability of diatoms to bioaccumulate rare earth elements by the example of lanthanum, one of the most abundant rare earth elements in the marine waters, for two strains of diatoms. As a result, the bioadsorption of lanthanum on the surfaces of frustules and its hyperaccumulation in the biomass of diatoms was revealed. This metal had a significant effect on the growth of diatoms. The results obtained regarding the bioaccumulation of lanthanum by diatoms are important for the further development of methods to treat aquatic environments contaminated with rare earth elements and for the respective bioindication.

**Abstract:**

Bioaccumulation by diatoms, as the first stage of biomineralisation, has been widely studied for various metals, such as cadmium, copper, zinc, aluminium, gold, silver, etc. However, despite the fact that the mining and utilization of rare earth elements (REEs) are currently increasing, there is almost no data on their bioaccumulation by diatoms. Therefore, the aim of this study was to determine the ability of diatoms to bioaccumulate REEs by the example of lanthanum (La), and to compare this ability for two marine diatoms *Nanofrustulum shiloi* and *Halamphora kolbei*. As a result of experiments on the cultivation of diatoms in nutrient media supplemented with La at concentrations of 10 mg·L^−1^ and 50 mg·L^−1^, energy-dispersive X-ray spectroscopy revealed the ability of diatoms to bioadsorb La on their frustule surface, as a first stage of its bioaccumulation. The high concentration of La (50 mg·L^−1^) has a noticeable visual effect on the morphofunctional state of diatoms and causes a decrease in the rate of accumulation. The low concentration (10 mg·L^−1^) promotes the hyperaccumulation of La by the diatom biomass as a whole, including both bioadsorption and bioabsorption within the cells. This resulted in an increase in La concentration in the biomass by nearly 2000-fold in *H. kolbei* (6.06 mg·g^−1^) and by 1000-fold in *N. shiloi* (6.90 mg·g^−1^). The results on La bioaccumulation by diatoms are significant for advancing methods to remediate aquatic environments contaminated with rare earth elements and for the bioindication purposes.

## 1. Introduction

In the marine environment, diatoms are among the dominant groups of algae in both phytoplankton and benthic communities [1,2]. They are the oldest and one of the most evolutionarily developed groups of microalgae. These organisms have a variety of adaptive mechanisms to cope with diverse environmental conditions and factors, which explains their ability to thrive in nearly any environment: from fresh to hypersaline waters, from Arctic ice to thermal springs, and even in soils. One of the adaptive mechanisms in diatoms is biomineralisation. The most notable example of this process is the transformation by their cells of an aqueous solution of silicic acid into silica to build the frustules—the hard and porous external cell wall of diatoms [3]. This unique process gives rise to the various complex frustule ornaments typical of each diatom species. The biomineralisation of iron by diatoms has also been studied in detail [4]. This element is crucial for photosynthesis. Since iron is present in seawater in very low concentrations (nanogram scale and below), diatoms have a number of adaptive strategies for iron accumulation and utilization [5]. Their cells contain proteins responsible for iron uptake and storage, as well as for providing defense against reactive oxygen species [6].

The first stage of biomineralisation in diatom cells is the accumulation of chemical elements—bioaccumulation. Bioaccumulation consists of three stages: (1) adsorption of the element onto the surface of the frustules (bioadsorption); (2) elemental transport into the cell via proteins; and (3) absorption and accumulation of the element inside the cell [7].

A significant amount of literature has documented the bioaccumulation of various metals by diatoms. The effects of heavy metals on diatoms have been extensively studied because these metals are common pollutants in aquatic environments [8,9]. Among the heavy metals, cadmium bioaccumulation has been the most thoroughly explored [7,10,11,12]. In addition, there are works on the bioaccumulation of zinc and copper [13], lead [14] and mercury [15]. There is also evidence of the accumulation of titanium [16], chromium [17], barium [18], aluminium [19], gold [20], silver [21] and germanium [22] by diatoms.

Nowadays, due to the massive production of high-tech devices and the rapid advancement of electronics and other advanced technologies, there is a boost in the use and extraction of rare earth elements (REEs). This results in their increased release into the environment, contributing to the formation of pollution hotspots that extend into seawater. For example, anthropogenic pollution of the Rhine River with lanthanum was detected in the densely populated city of Worms, reaching a concentration of 49 mg/L, while near the cracking catalyst producing plant, Ce concentration reached 2.5 mg/L [23]. Despite the fact that at the Crimean coast of the Black Sea there is no mining and processing of REEs, long-term monitoring (1990–2002) of the elemental composition of coastal waters of the western and southern coasts of Crimea revealed the elevated REE concentrations decreasing in the following order: lanthanum (La, 0.12–0.22 nM) > cerium (Ce, 0.05–0.15 nM) > scandium (Sc, 0.028–0.058 nM) > terbium (Tb, 0.0058–0.0066 nM) > europium (Eu, 0.0026–0.0046 nM) > lutetium (Lu, 0.0013–0.0024 nM) > ytterbium (Yb, 0.0021–0.0022 nM) > samarium (Sm, 0.0008–0.0018 nM) [24,25]. In addition, it was shown that atmospheric precipitation in the area of Sevastopol in 2008–2009 was a significant source of pollution with the following REEs: scandium (Sc), lanthanum (La), cerium (Ce), samarium (Sm), europium (Eu), terbium (Tb), neodymium (Nd), ytterbium (Yb), and lutetium (Lu) [26]. According to the more recent data, all REEs except promethium have been detected in water and bottom sediments of the southern and southwestern coast of Crimea [27]. In that study, the highest concentration in seawater was observed for scandium (51.4–51.8 ng·L^−1^) and lanthanum (up to 15 ng·L^−1^).

However, existing data on the REE bioaccumulation by diatoms are sporadic and not exhaustive [28,29,30]. The toxic effect of lanthanides on the planktonic diatom *Skeletonema costatum* was revealed in [31]. Recently, the biomineralisation ability of diatoms has been used to synthesise lanthanum and yttrium nanoparticles, whereupon the frustules with embedded nanoparticles were used as an anode material for lithium-ion batteries to increase their capacity [32]. The referenced studies do not provide direct evidence of bioaccumulation of REE by living diatom cells. Instead, they offer only indirect evidence or focus on dead diatom frustules. In our study, we aimed to test the hypothesis that living diatom cells can accumulate REEs. Therefore, the goal of this study was to determine the ability of diatoms to bioaccumulate REEs by the example of lanthanum, one of the most abundant REEs in the Crimean coastal waters, and to compare this ability for two marine diatoms *Nanofrustulum shiloi* (J.J.Lee, Reimer et McEnery) Round, Hallsteinsen et Paasche 1999 and *H. kolbei* (Aleem) Álvarez-Blanco et S. Blanco 2014.

## 2. Materials and Methods

### 2.1. Isolation of Diatoms and Strain Characteristics

Two strains of diatoms (Bacillariophyta), *N. shiloi* DIA-N.sh.-KarBay and *H. kolbei* DIA-A.big.-SevBay, from the microalgal and cyanobacterial collection of the Department of Aquaculture and Marine Pharmacology at A.O. Kovalevsky Institute of Biology of the Southern Seas of RAS were used to study the bioaccumulation of lanthanum.

To isolate these strains, mixed cultures of diatoms obtained from benthic environments were used. Two methods were employed: multiple passages using Pasteur pipettes and micromanipulation techniques. The multiple passages method involved inoculating 1 mL of enriched culture into Petri dishes containing 15–20 mL of F/2 medium, followed by several transfers to maintain sterility. In the micromanipulation method, individual cells were transferred using Pasteur pipettes to a drop of nutrient medium on a glass slide under a microscope and then moved to Petri dishes. The isolated strains were examined under a light microscope (LM), Axioskop 40 C. Zeiss (Jena, Germany). For more precise identification of diatoms, scanning electron microscopy (SEM; Hitachi SU3500, Tokyo, Japan) was employed. Samples for SEM imaging were prepared following Simonsen’s method [33].

*N. shiloi* is a marine and freshwater benthoplanktonic colonial diatom (Figure 1). This species can live in plankton due to the structure of its frustules, which have spines. It forms colonies shaped as long chains that can settle on the bottom under their own weight. *N. shiloi* was isolated from Karantinnaya Bay (Crimea, Black Sea) [34]. *N. shiloi* is penate and seamless; its cells are cylindrical in shape. Frustules have a rectangular height of 2.6–3.2 µm. The valve surface is subcircular, flat or slightly domed, 2.8–3.6 µm in diameter. For this species, identification was carried out, and the biochemical composition was studied [34,35]. *N. shiloi* is a promising object in biotechnology [35].

*H. kolbei* was isolated from the benthos of the northern coast of Sevastopol Bay (Crimea, Black Sea) (Figure 2). The cells are small, but bigger than those of *N. shiloi*: 8.6–15 µm long, 3.5–8.5 µm wide in valve view and up to 12 µm in gridle view. Frustules are elliptical, constricted in the middle. On the dorsal side, the species has numerous gridle bands. Chloroplast is single H-shaped. In culture, the species forms a film. The studies [36,37,38] were used to identify the species.

Axenic cultures of these algae were grown in conical flasks with sterile medium F/2 [39] prepared using filtered seawater containing of the following nutrients: 150 g·L^−1^ NaNO_3_, 10 g·L^−1^ NaH_2_PO_4_ × H_2_O, 60 g·L^−1^ Na_2_SiO_3_ × 9H_2_O, 0.0196 g·L^−1^ CuSO_4_ × 5H_2_O, 0.044 g·L^−1^ ZnSO_4_ × 7H_2_O, 0.02 g·L^−1^ CoCl_2_ × 6H_2_O, 0.36 g·L^−1^ MnCl_2_ × 4H_2_O, 0.0126 g·L^−1^ Na_2_MoO_4_ × 2H_2_O, 8.7 g·L^−1^ Na_2_EDTA, 6.3 g·L^−1^ FeSO_4_ × 7H_2_O.

Purity of the cultures was maintained by Gentamicin (40 mg·mL^−1^ at a dose of 2 mL per litre of medium). The culture media were sterilised in the autoclave Tuttnauer 2540 ML (Tel-Aviv, Israel) at a temperature of 120 °C and a pressure of 1 atm within 20 min. The cultivation was conducted in the climatic chamber KS-200 (Smolensk, Russia) at a temperature of 18 °C and a 10 h/14 h light/dark cycle.

### 2.2. Cultivation of Diatoms in Lanthanum-Supplemented Medium

The two diatom strains were cultivated in the F/2 nutrient medium with the addition of La at low (10 mg·L^−1^) and high (50 mg·L^−1^) concentrations (hereinafter “experimental cultures”). Such concentrations fall within the typical range for studying the effect of metals on diatom algae (e.g., for cadmium [7]). Our study was based on the previous studies of two species of diatoms [40] and microalgae *Raphidocelis subcapitata* [41], in which the concentration of 10 mg·L^−1^ inhibited growth of the diatoms to a small extent and practically did not inhibit *R. subcapitata*. This information enables setting up a long-term experiment. The concentration of 50 mg·L^−1^was assumed to be potentially toxic to diatoms.

To obtain the above concentrations, lanthanum (III) nitrate hexahydrate (La(NO_3_)_3_·6H_2_O) was dissolved with EDTA in the sterile medium F/2. Then, the solution was pasteurised. All reagents used in the culture media preparation were purchased from Lenreaktiv (St. Petersburg, Russia) and were of analytical grade. The pH of the prepared solutions at a La concentration of 10 mg·L^−1^ was 7.01, and at 50 mg·L^−1^ it was 6.68. One milliliter of inoculum from each species was transferred into sterile Petri dishes containing La-supplemented medium. Control conditions were the same, but without adding La. The inoculum density was in the range of tens of thousands of cells·mL^−1^. The experiments were performed in three replicates under the same conditions. Samples for further analyses from experimental and control cultures were taken after 1, 3, 6 h, then after 1, 3, 6, 9 days and also after 12, 16 days at low La concentration and after 14, 20 days at high La concentration. For the abundance counting, cultures were thoroughly mixed throughout the entire volume of the dish using a sterile glass rod, and a 1.5 mL aliquot was taken. The aliquot was placed in microtubes and fixed with 96% ethanol for further processing.

For the element analysis in the biomass, diatoms were grown for one month in sterile 1 L flasks in the F/2 nutrient media with the addition of La at a concentration of 10 mg·L^−1^. Cultures were maintained at 18 °C under natural light and aerated using an aquarium compressor. At the end of the cultivation period, the density of the cultures was estimated by direct biomass weighing. In this procedure, diatom cells were first precipitated by centrifugation at 1600× *g* for 10 min.

### 2.3. Sample Processing and Microscopy

Visual control of the state of the diatom cultures and microphotographing were carried out under a LM Axioskop 40 C. Zeiss and using SEM, Hitachi SU3500. To estimate the abundance of the diatoms, cells were counted in a hemocytometer (Goryaev’s chamber, LLC Minimed, Bryansk, Russian Federation) with a volume of 0.9 mm^3^ under the LM at the 10 × 40 and 10 × 100 magnifications. The counting was performed in triplicate for each sample. The results were then converted to a per milliliter basis. The SEM was used to study the structure of the surfaces of diatom frustules in detail. To this end, their samples were subjected to rapid drying in a critical point drier (EM CPD300, Leica, Wetzlar, Germany) using CO_2_. Then, to provide electrical conductance, the surface of the samples was coated with a thin layer of gold-palladium alloy using an ion sputter coater ACE200 (Leica, Germany) to further analyze by SEM. For each valve, an EDX spectrum was recorded separately.

Energy dispersive X-ray spectroscopy (EDX) was used to determine the elemental composition of diatom frustule surfaces and La content (bioadsorbtion). Samples from the control and experimental cultures were collected with a sterile pipette. One drop from each dish was placed on the cover glass and then rinsed repeatedly with distilled water using an automatic pipette. EDX spectra were obtained using of an Oxford Instruments Ultim Max detector (Abingdon, UK) integrated into SEM. For each sample, 10–20 diatom frustules were analysed. As a result of the analysis, the elemental concentrations were determined in weight %. To eliminate the influence of the gold-palladium coating applied for preparing the samples for SEM, the Au/Pd data were subtracted from the EDX spectra at the software level. When analyzing the elemental content using EDX, the analysis depth was on the order of several micrometers. It is important to note that EDX analysis is a technique suitable only for surface probing, and its results reflect the elemental composition of the surface layer of diatom frustules.

### 2.4. Analytical Sample Preparation and Elemental Analysis in Diatom Biomass

To analyze the bioabsorption of La by diatom biomass as a whole, including bioadsorption and bioabsorption into the cells, after weighing the control and experimental cultures grown over the month, their biomass precipitated via centrifugation was washed twice with isotonic sodium chloride solution (9 g·L^−1^) to remove residues of the nutrient medium and lanthanum. After washing, the biomass was dried in a RA-50/500 VZ drying oven (Micron, Shatura, Russia) at 105 °C for 24 h. Then, samples of dry homogenized diatom biomass (200 mg) were weighed on an analytical balance with a precision of 0.1 mg and placed in PTFE digestion vessels equipped with screw caps. Then, 3 mL of hydrofluoric acid (70%) was added to the biomass samples, and the vessels were kept closed on a glass-ceramic hotplate at 50 °C for about 2 h. The excess acid was evaporated from the vessels kept in a sand bath at 150 °C until a wet residue was obtained. After this procedure, 3 mL of a mixture of extra pure concentrated nitric and hydrochloric acids (3:1), preliminarily purified by subboiling distillation, was added and maintained at 180 °C until the solid residue was completely dissolved. Then, 10 mL of bidistilled water was added and the solution was left for one hour. The sample solution was cooled, filtered through a paper filter and diluted to 50 mL in a measuring flask. In parallel, blank samples were prepared, and the element concentrations in them were subtracted from the element concentrations in the biomass samples.

Concentrations of macro- and trace elements were measured using inductively coupled plasma mass spectrometry (ICP-MS) on an instrument Agilent 7700x (Santa-Clara, CA, USA). The measurement mode was set in accordance with the manufacturer’s recommendations. The matrix effect and instrumental drift of the instrument baseline were corrected using a solution of indium (200 μg·mL^−1^) as an internal standard. Calibration solutions were prepared from multielement standard solutions IV-ICPMS-71A–D (Inorganic Ventures, Christiansburg, VA, USA) with certified element concentrations (10 mg·L^−1^). The relative error of determination in the working ranges of element concentrations did not exceed 7%. The quality control of the analysis was warranted by measuring element concentrations in the standard reference material SRM^®^ 2976 “Mussel Tissue” (National Institute of Standards & Technology, Gaithersburg, MD, USA). The recoveries of the concentrations of all elements were within 85–123%. The analytical accuracy was within 10%. Element concentrations in samples are expressed on a dry weight basis.

### 2.5. Statistical Analysis

All experiments were performed in triplicate (*n* = 3), and all data are presented as mean ± 95% confidence interval. Prior to the statistical analysis, the Shapiro–Wilk and Levene’s tests were performed to ensure normality and homogeneity assumptions. PAST 4.17 [42] statistics software was employed to perform the statistical analysis, which included two-way PERMANOVA at *p* ≤ 0.05. MS Excel 2021 software was used for calculations. OriginPro 2024 SR1 10.1.0.178 software was used to create plots.

## 3. Results

### 3.1. Nanofrustulum shiloi and Halamphora kolbei Cultivation in Media Supplemented with La

To assess the bioaccumulation of La by diatoms and to study its effect on the microalgal cultures, strains of *Nanofrustulum shiloi* and *Halamphora kolbei* were cultivated after the addition of two concentrations of La into the nutrient media: low (10 mg·L^−1^) and high (50 mg·L^−1^).

#### 3.1.1. La Concentration 10 Mg·L^−1^

Visual monitoring. During the experiment, there was no significant difference in the cultural characteristics and morphophysiological state of diatom cells in La-supplemented and control media (Figure 3).

Examination under a light microscope revealed that in the La-supplemented medium, frustules of both strains retained their shape, chloroplasts had a rich unchanged color and cells appeared natural (Figure 4).

Abundance estimation. The dynamics of the abundance of both cultures (Figure 5 and Figure 6) generally reflect the trends observed in the visual assessment, demonstrating their well-being. However, the PERMANOVA test showed significant differences between the abundance of the experimental and control cultures (*p* < 0.01) (See Appendix A). Throughout the experiment, the abundance of diatoms in the La-supplemented and control medium increased, reaching the values of 9111 ± 694 thousand cells·mL^−1^ and 10,864 ± 883 thousand cells·mL^−1^, respectively, for *N. shiloi* 230 ± 53 thousand cells·mL^−1^ and 269 ± 32 thousand cells·mL^−1^, respectively, for *H. kolbei*. The growth dynamics of *N. shiloi* in a medium containing La were characterised by a prolonged lag phase up to one day, followed by accelerated growth up to six days, reaching control values. The control culture continued to grow until the end of the exposure period (Figure 5). In contrast, the experimental culture exhibited a slight decrease in the abundance, followed by its increase.

The abundance of *H. kolbei* in the experimental culture was lower than in the control (Figure 6, Appendix A). Up to three days of cultivation, both cultures of *H. kolbei* developed synchronously, after which the abundance of the experimental culture decreased significantly until day 9, while in the control, it continued to grow. By day 12, the abundance values of both cultures increased sharply, and their dynamics became synchronised. On day 16, the abundance values of *H. kolbei* in the La-supplemented medium reached the control level.

SEM and EDX analysis. Overall, the La concentration on the individual frustule surfaces varied significantly and ranged from 0 to 4.8% for *N. shiloi* and from 1.9 to 10.2% for *H. kolbei* throughout the entire exposure period. Examples of EDX spectra for individual frustules of the diatoms are shown in Figure 7.

During the first day of exposure, the average La content increased from 0.11 to 0.73% for *N. shiloi* and from 0.25 to 3.02% for *H. kolbei* (Figure 8). Moreover, after the first hour of exposure, its content for individual frustules was 0–0.4% for *N. shiloi* and 0.2–0.3% for *H. kolbei* (Appendix A). The median values for the two species increased, reaching a maximum on day 3 (med = 0.8 Wt. % for *N. shiloi* and med = 4.0 Wt. % for *H. kolbei*). Further, with an increase in the exposure period from day 6 to day 12, the La content decreased. Thus, the median lines in Figure 8 illustrate the trend of La bioaccumulation described above: at the beginning of the experiment, the metal content on the surface of the frustules of both species increases, reaching maximum levels on the third day, and then decreases. This is especially noticeable for *H. kolbei*.

#### 3.1.2. La Concentration 50 Mg·L^−1^

Visual monitoring. In media with high La concentrations, by the third day, cultural characteristics differed between the experimental and control cultures, with the latter developing more actively (Figure 9). By the ninth day, the *N. shiloi* control culture had almost completely covered the bottom of the Petri dishes, and biomass was clearly visible, unlike in the medium with La. *H. kolbei* demonstrated slower growth by the third day. Under control conditions, the biomass was clearly visible; however, in media containing La, the biomass was not visible to the unaided eye. By the ninth day, *H. kolbei* growth was clearly visible in the control Petri dishes, but not in the La-containing media.

When assessing the morphophysiological state of the cultures under a LM, the difference in the *N. shiloi* strain on day 3 is almost imperceptible. However, by day 9, the chloroplasts exhibit a paler color and a lower density compared to the control (Figure 10). For *H. kolbei*, chloroplasts by day 3 were already severely deformed and had changed color. By day 9, the culture contained empty frustules, and the chloroplasts were almost completely discolored.

Abundance estimation. At a high concentration of La in the medium, the abundance of the two species in experimental cultures was significantly lower than in the control throughout the whole period of exposure (*p* < 0.0001) (Appendix A). The abundance of *N. shiloi* (Figure 11) in the experimental cultures gradually increased during the first six days, in contrast to the control culture, where the growth was erratic. Further, its abundance in the La-supplemented medium remained at a relatively constant level until day 20 and beyond. *H. kolbei* grew during the first three days, after which its abundance in the experimental cultures remained approximately constant until day 14, when it declined and eventually died, unlike the control cultures, which continued to grow (Figure 12).

SEM and EDX analysis. Lanthanum was absent on the diatom frustule surfaces during the first 6 h. After three days, the La content did not exceed 0.2 Wt. % (Appendix A, Figure 13). Further, the metal content on *N. shiloi* frustules increased, reaching a maximum on day 9, after which it decreased for *N. shiloi*, while the *H. kolbei* culture died. Overall, the La content on the surfaces of individual frustules varied from 0.7 to 3.5 Wt. % for *N. shiloi* and from 0 to 0.2% for *H. kolbei* (Appendix A). Moreover, throughout the experiment, *H. kolbei* mainly had frustules on which no La was detected (Appendix A, Figure 13).

Thus, at high concentrations of La in nutrient media, its bioaccumulation on frustule surfaces occurs more slowly and to lower values than in nutrient media with low La concentrations.

### 3.2. Elemental Composition of the Biomass of Nanofrustulum shiloi and Halamphora kolbei Cultivated in the 10 Mg·L^−1^ La-Supplemented and Control Media

The elemental analysis of the biomass of the two diatom species involved quantitation of 45 elements (Appendix A). Hyperaccumulation of La was detected in both experimental diatom cultures on the basis of diatom biomass as a whole, including bioadsorption on the frustule surfaces and bioabsorption within the cells. This was manifested in La concentrations of 6065.5 μg·g^−1^ and 6905.9 μg·g^−1^ in *H. kolbei*. and *N. shiloi* biomass, respectively. Lanthanum was also found in the control biomass samples, as the culture media were prepared using natural seawater. However, in the control, its concentrations were significantly lower than in the La-containing media: 3.084 μg·g^−1^ for *H. kolbei* and 7.072 μg·g^−1^ for *N. shiloi*. But even such lanthanum concentrations in the biomass exceed the concentration in the seawater of the sampling area (0.015 μg·L^−1^ [27]) by 206 times for *H. kolbei* and by 471 times for *N. shiloi*. The content of La in the biomass and on the frustule surfaces of *H. kolbei*, cultivated in a medium containing 10 mg La·L^−1^, is higher than that in *N. shiloi*. *H. kolbei* accumulated greater amounts of La; however, it proved to be more sensitive to it, as the strain did not survive at high La concentrations in the medium. In addition to lanthanum, rare earth elements such as cerium, praseodymium, neodymium, samarium, europium, gadolinium, dysprosium, holmium, erbium, thulium, ytterbium, and lutetium were found in the biomass of the two strains, both in the experimental and control cultures, but at lower concentrations, below 1 μg·g^−1^ (Appendix A).

The effect of lanthanum on diatom physiology is evident from the elemental level assessment. The two strains studied responded similarly to the prolonged influences of the metal, but to varying degrees with respect to accumulating individual elements. A decrease in the concentration of such elements as sodium, phosphorus, sulfur, potassium, calcium, manganese, strontium, molybdenum and barium in the biomass content was observed, alongside an increase in the content of iron, nickel, copper and selenium. Both strains exhibited a similar decrease in the phosphorus, calcium, and barium content, as well as an increase in the iron and selenium content. The range of changes in the concentrations of sodium, potassium, nickel, and selenium was greater for *H. kolbei* than for *N. shiloi*, indicating more profound disturbances in its metabolic processes. These disturbances were revealed by both visual assessment and abundance dynamics.

## 4. Discussion

The study of the effects of La on microorganisms is a fairly widely researched topic in experimental science [43,44,45,46,47,48,49]. For example, cyanobacteria have already been proven to be capable of biosorbing lanthanides. In these microorganisms, the lanthanide biosorption begins immediately, and within the first few minutes of exposure, the metal is almost completely eliminated from the solution [50,51]. For *Arthrospira platensis*, a high affinity of biomass for La was observed, with the metal accumulation ranging from 15.99 to 39.8 mg·g^−1^ [52]—an order of magnitude higher than the values obtained in this work. However, as mentioned above, such studies are rare in relation to diatoms, and the La bioaccumulation has not been studied at all.

As a result of our experiments on the exposure of diatom strains *N. shiloi* DIA-N.sh.-KarBay and *H. kolbei* DIA-A.big.-SevBay to media supplemented with low (10 mg·L^−1^) and high (50 mg·L^−1^) concentrations of La, the process of La bioadsorption on the surface of frustules was studied for the first time. The highest metal concentration was observed in media with the low La concentrations, while in media with the high concentrations of La, its accumulation was low throughout the entire exposure period. It is worthwhile to note that for *H. kolbei*, at a low La concentration, no frustules with zero La content were observed in most cases, while for *N. shiloi*, frustules with no La on them were found almost throughout the entire experiment. This is probably due to the higher division rate of *N. shiloi*, which results in a larger number of nascent cells that have not had enough time to adsorb the metal from the medium.

Visual monitoring of the morphophysiological state of the cultures with the addition at a low concentration of La in the medium did not reveal any significant changes. In contrast, a high concentration of La adversely affected the state of the diatoms. This effect was particularly pronounced in *H. kolbei*, whose state had significantly deteriorated by day 3, and by day 14, the culture had completely died. Therefore, a La concentration of 50 mg·L^−1^ in the medium is lethal for *H. kolbei*. At the same time, *N. shiloi* proved to be resistant to this concentration, as its cells remained viable for 20 days of exposure and longer.

The dynamics of the strain abundance in both the experimental and control cultures are similar, but they differ in the timing of the growth phase onset. In the La-supplemented media, the cultures required an initial adaptation period of one day, after which more active growth was observed. This effect is particularly evident for *N. shiloi* cultivated in a medium containing 10 mg La·L^−1^ (Figure 5). The strain then exhibited active growth until day 6, reaching levels comparable to the control. Subsequently, similar to the control, the negative effects of accumulated secondary metabolites, along with the continued presence of lanthanum, caused a noticeable decline in abundance up to day 12. However, the culture eventually adapted and resumed growth.

When comparing the abundances of each diatom species in the control and experimental groups at both high and low La concentrations in the media (using two-way PERMANOVA), statistically significant differences were observed throughout the experiment (Appendix A). This confirms that lanthanum affects diatom growth even at moderate concentrations. Growth inhibition by lanthanides, including La, has also been demonstrated in the planktonic diatom *Skeletonema costatum*, both for individual metals and in mixed solutions [31]. For two species of diatoms, lanthanum did not have an inhibitory effect at concentrations of 0.1 and 1 mg·L^−1^ and even had a slight stimulating effect on the pigment production. However, at 10 mg·L^−1^, La caused oxidative stress, disrupting the antioxidant system of diatoms [40].

In addition to the La hyperaccumulation in the diatom biomass as a whole, upon the La addition to the nutrient medium, other changes in the elemental composition of the biomass of the diatoms were also observed. For example, decreases in the concentrations of sodium, potassium and calcium in the biomass may be primarily attributed to increased permeability of cell membranes, resulting in the uncontrolled release of these essential ions from the cell against their concentration gradients. The loss of potassium, which is critically important for osmoregulation and enzyme function, occurs first [53]. The concentration of this element in the biomass decreased by 3.6-fold in *H. kolbei* and by 1.2-fold in *N. shiloi* compared to the control. Lanthanum is a known calcium channel blocker [54]; its ions substitute for calcium, thereby preventing its entry. It is also noteworthy that the concentrations of strontium and barium—chemical analogues of calcium—decreased in the biomass, likely because their membrane entry was similarly impaired. Upon the addition of La, the concentration of strontium increased, while the concentration of calcium decreased. It is likely that strontium accumulates in diatoms by replacing calcium, as its ions have a radius close to that of calcium ions. Also, the addition of La results in a twofold increase in iron concentration in the biomass of *H. kolbei* and in a 1.5-fold increase in *N. shiloi*, which may also represent a stress response by the organisms.

A decrease in the phosphorus and sulfur content—key components of biomolecules—in the biomass of experimental cultures by factors of 2.4–2.5 and 1.4–1.8, respectively, indicates a profound suppression of cellular metabolism. It is known that rare earth elements form strong, insoluble complexes with phosphate ions in the medium [55], rendering them inaccessible for cellular uptake. Manganese and molybdenum are essential metals in enzymes involved in the photosystem II functioning and nitrogen metabolism [56]. The manganese concentration in the biomass decreased 4.1-fold for *N. shiloi* and 5.0-fold for *H. kolbei*, while molybdenum, an essential trace element for diatoms [56], increased by 1.6 times in *N. shiloi* but decreased by 0.6 times in *H. kolbei*. Iron is known to act as a cofactor in molybdenum assimilation. In our experiment, both diatom species exhibited increased concentrations of elements such as iron, nickel, and copper, indicating activation of antioxidant enzymes (catalase, peroxidase, superoxide dismutase, including Ni-dependent isoforms) [57,58] in response to the intense oxidative stress induced by the exposure to lanthanum [41]. Significant increases in the selenium concentration were observed in the biomass of both strains. In *H. kolbei*, selenium concentrations increased 581-fold, while in *N. shiloi*, the increase was 1322-fold. For some diatoms, selenium is an essential element for growth in culture and can enhance resistance to stress conditions, as it is a component of selenoproteins and enzymes such as glutathione peroxidase, which protect cells from oxidative damage [59,60]. These findings suggest that *N. shiloi* exhibits greater metabolic flexibility and adaptive capacity, whereas *H. kolbei* demonstrates higher sensitivity to toxicants.

Notably, 13 rare earth elements were detected in both experimental and control samples grown on a nutrient medium prepared from filtered seawater without any added La. All these elements were found in the water near the sampling site [27]. Among the elements found in the biomass, the highest concentrations were observed for La (including control cultures) and cerium, in agreement with the data on seawater, in which the concentration of lanthanum varied from 0.001 to 0.15 μg·L^−1^, and cerium from 0.001 to 0.011 μg·L^−1^. The concentrations of other rare earth elements in the water were significantly lower, on the order of several ng·L^−1^ and below. This indicates the ability of diatoms to bioaccumulate rare earth elements even at low concentrations in the environment. Since the media were prepared using natural seawater, it can be assumed that the same process will occur in the marine environment. However, this assumption requires further research.

It is interesting to compare the results of studies on the bioaccumulation of lanthanum and other metals. For example, Schmitt et al. [14] studied the influence of several heavy metals—zinc, lead, copper, cadmium, mercury—and aluminum on two planktonic diatom species using ICP-MS. The concentrations of these metals in the biomass of *Stephanocyclus cryptica* ranged from 11.9 mg·g^−1^ dry weight (mercury) to 243.0 mg·g^−1^ (zinc), whereas in the biomass of *Phaeodactylum tricornutum*, they ranged from 0.5 mg·g^−1^ dry weight (mercury) to 14.5 mg·g^−1^ (zinc). Thus, the two species responded differently to the addition of metals in the nutrient medium, resulting in varying concentrations of the accumulated metals in their biomass. In the biomass of *S. cryptica*, the concentration of zinc was two orders of magnitude higher, and the concentrations of other metals were an order of magnitude higher than those we observed for La. In *P. tricornutum*, the zinc concentration was an order of magnitude higher, while the concentrations of the other metals were of the same order of magnitude as those recorded for La. In the freshwater species *Eolimna minima*, gold bioaccumulation was studied using ICP-MS [20]. The concentration of gold in the biomass exceeded the La values by an average of fourfold, reaching 24.6 mg·g^−1^. In another study [61], the concentration of gold measured by Raman spectroscopy in the biomass of the marine planktonic diatom *Eupyxidicula turris* reached as much as 80–105 mg·g^−1^. Close to the La concentrations were concentrations of cadmium determined using FTIR spectroscopy (1.23–5.22 mg·g^−1^) [11]. However, in [13] the concentration of cadmium measured using atomic absorption spectrometry in the biomass of the freshwater diatom *Craticula subminuscula* reached 42 mg·g^−1^, copper reached 90 mg·g^−1^, and the highest bioaccumulation was observed for zinc at 127 mg·g^−1^, consistent with the findings reported in [14].

Finally, it should be noted that in the present study, preliminary data were obtained that indicate the ability of diatoms to bioaccumulate lanthanum. However, this initial result requires further detailed research using a wider range of concentrations and a larger number of strains with a control of the initial and final metal concentration in the nutrient medium to calculate the total sorption capacity of diatoms. It is necessary to test the ability of diatoms to bioaccumulate other rare earth elements, including non-lanthanides. The knowledge gained in this research can be used to develop technologies for the bioremediation of water environments contaminated with REEs and for the REEs pollution bioindication. In addition, the ability of diatoms to bioaccumulate and biomineralise rare earth elements can be used to obtain nanoparticles of these elements [40], which can be used to improve lithium-ion battery technologies [32] and create new composite materials [62] and UV filters [30].

## 5. Conclusions

The ability of diatoms to bioaccumulate lanthanum has been experimentally demonstrated, and this metal has been shown to have a significant impact on the growth of the species. The high concentration (50 mg·L^−1^) of La has a noticeable visual effect on the morphofunctional state of diatoms and causes a decrease in the rate of accumulation. The low concentration (10 mg·L^−1^) promotes the La hyperaccumulation by diatom biomass as a whole, including bioadsorption and bioabsorption within cells. It has been manifested in an increase in La concentration in the biomass by nearly 2000-fold in *Halamphora kolbei* (6.06 mg·g^−1^) and 1000-fold in *Nanofrustulum shiloi* (6.90 mg·g^−1^). Additionally, changes in the elemental composition of the diatom biomass were detected, indicating a clear species-specific response to the La supplementation.

The results on the La bioaccumulation by diatoms are significant for advancing methods to remediate aquatic environments contaminated with rare earth elements and for bioindication purposes.

## Figures and Tables

**Figure 1 biology-14-01489-f001:**
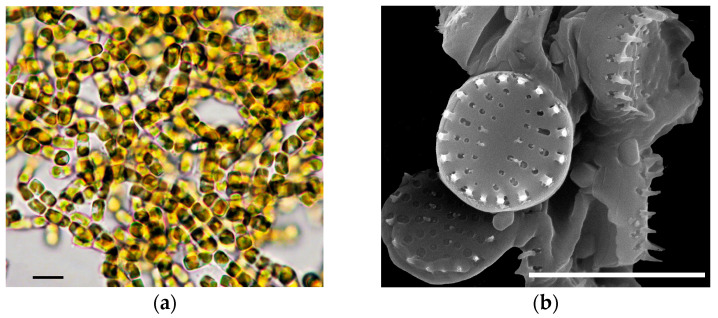
Diatom *Nanofrustulum shiloi* in culture: (**a**) living cells under a light microscope (LM) and (**b**) scanning electron microscopy (SEM) image of frustules. Scale bar: (**a**) 20 µm, (**b**) 5 µm.

**Figure 2 biology-14-01489-f002:**
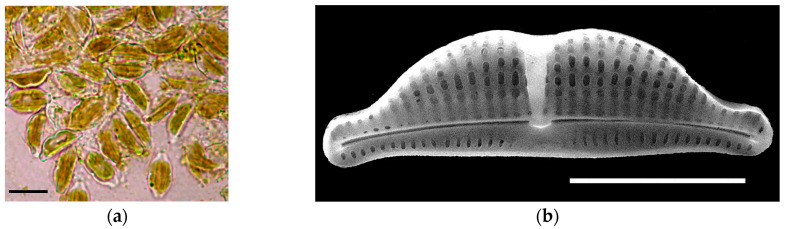
Diatom *Halamphora kolbei:* (**a**) living cells, LM, (**b**) SEM image of frustule. Scale bar: (**a**) 10 µm, (**b**) 5 µm.

**Figure 3 biology-14-01489-f003:**
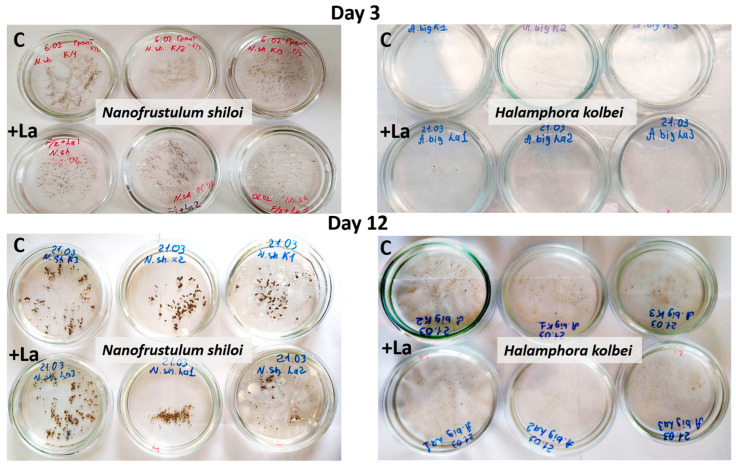
Visual assessment of the cultural characteristics and morphophysiological state of the diatom strains exposed to media containing 10 mg·L^−1^ La (+La) and without it (C) on days 3 and 12 of the experiment.

**Figure 4 biology-14-01489-f004:**
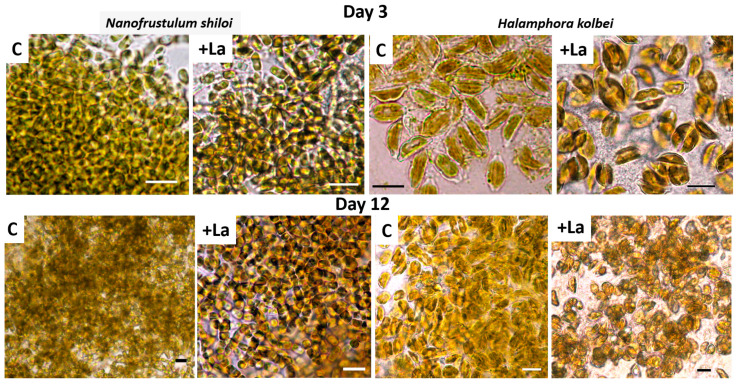
The morphophysiological state of the diatom cells under LM exposed to media containing 10 mg·L^−1^ La (+La) and without it (C) on days 3 and 12 of the experiment. Scale bar: 10 µm.

**Figure 5 biology-14-01489-f005:**
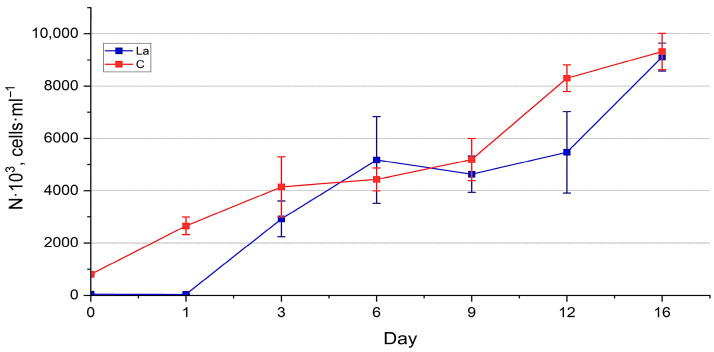
Dynamics of abundance of *Nanofrustulum shiloi* cultivated in the 10 mg·L^−1^ La-supplemented medium vs. control (C).

**Figure 6 biology-14-01489-f006:**
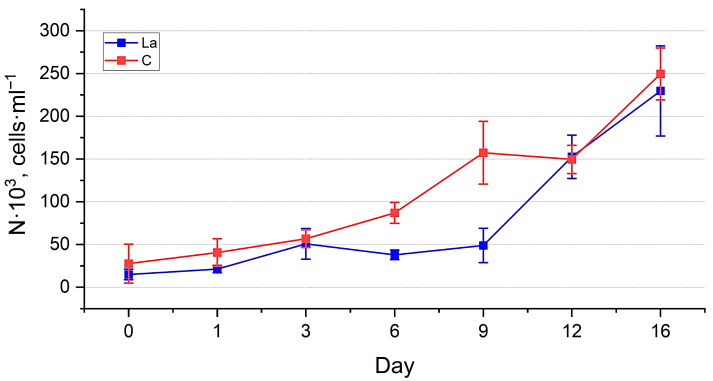
Dynamics of abundance of *Halamphora kolbei* cultivated in the 10 mg·L^−1^ La-supplemented medium vs. control (C).

**Figure 7 biology-14-01489-f007:**
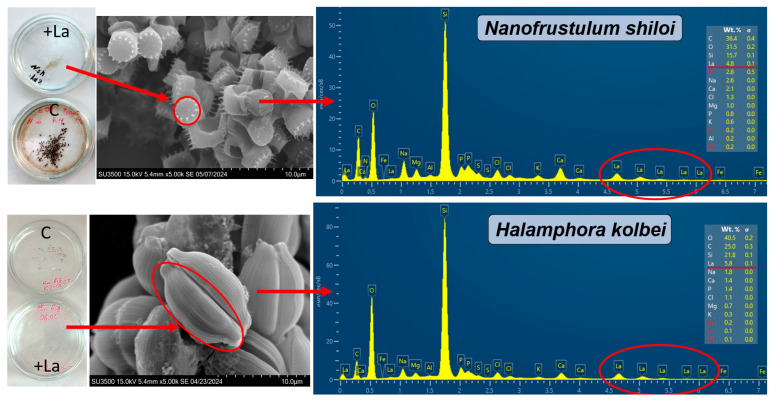
EDX spectra of the surface of diatom valves during cultivation at 10 mg La·L^−1^ in the medium. The red arrows show the scheme for obtaining the EDX spectra of the diatom valve surfaces. The red circles indicate La on the EDX spectra, and the lines indicate the percentage of La on the valve surfaces.

**Figure 8 biology-14-01489-f008:**
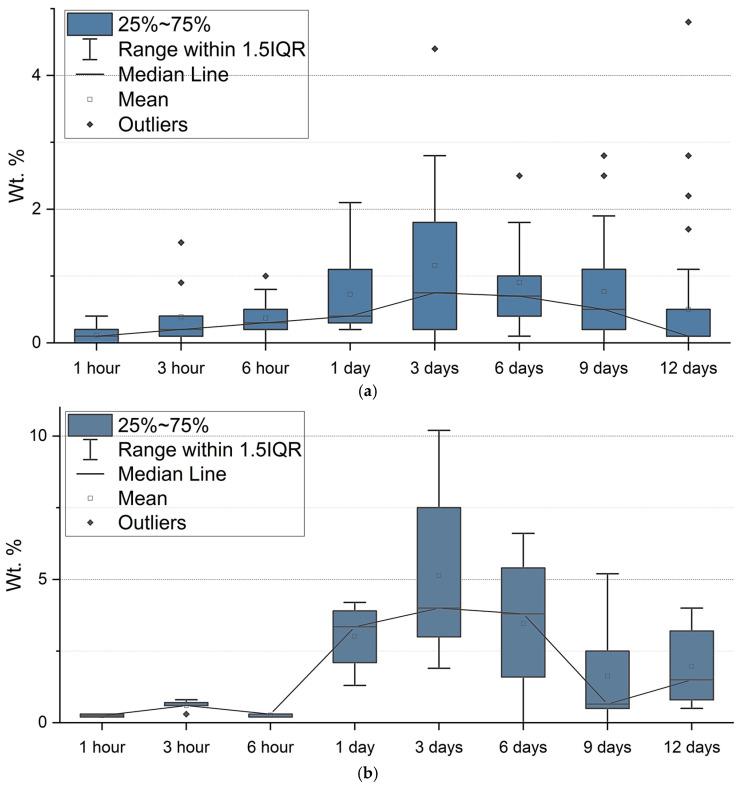
Weight fraction (%) of La on the frustule surfaces of (**a**) *N. shiloi* and (**b**) *H. kolbei* at 10 mg La·L^−1^ in the medium.

**Figure 9 biology-14-01489-f009:**
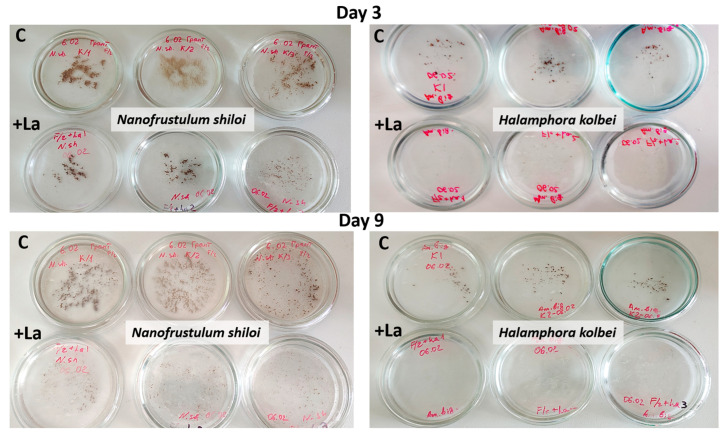
Visual assessment of the cultural characteristics and morphophysiological state of the diatom strains exposed to media containing 50 mg·L^−1^ La (+La) and without it (C) on days 3 and 9 of the experiment.

**Figure 10 biology-14-01489-f010:**
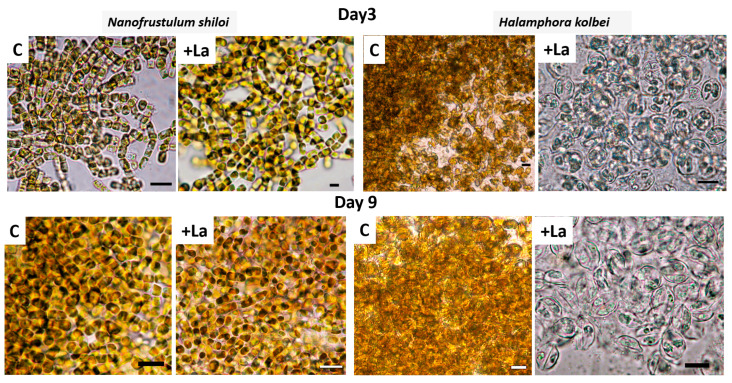
The morphophysiological state of the diatom cells under LM exposed to media containing 50 mg·L^−1^ La (+La) and without it (C) on days 3 and 9 of the experiment. Scale bar: 10 µm.

**Figure 11 biology-14-01489-f011:**
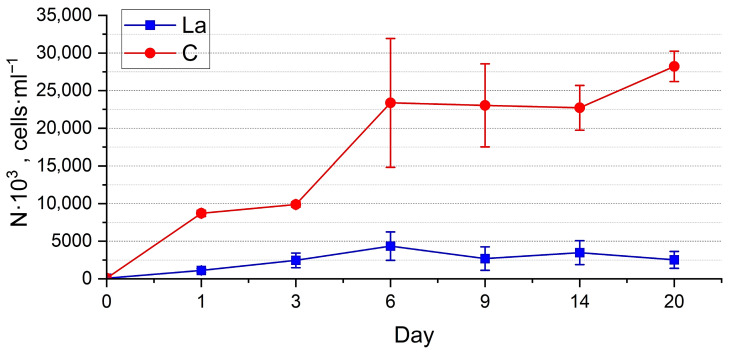
Dynamics of abundance of *Nanofrustulum shiloi* cultivated in the 50 mg·L^−1^ La-supplemented medium vs. control (C).

**Figure 12 biology-14-01489-f012:**
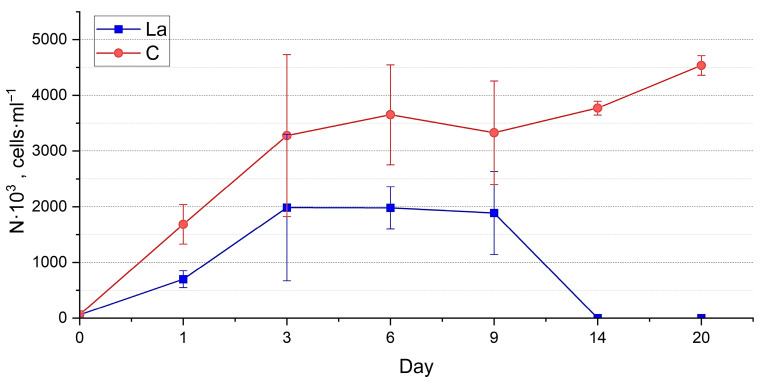
Dynamics of abundance of *Halamphora kolbei* cultivated in the 50 mg·L^−1^ La-supplemented medium vs. control (C).

**Figure 13 biology-14-01489-f013:**
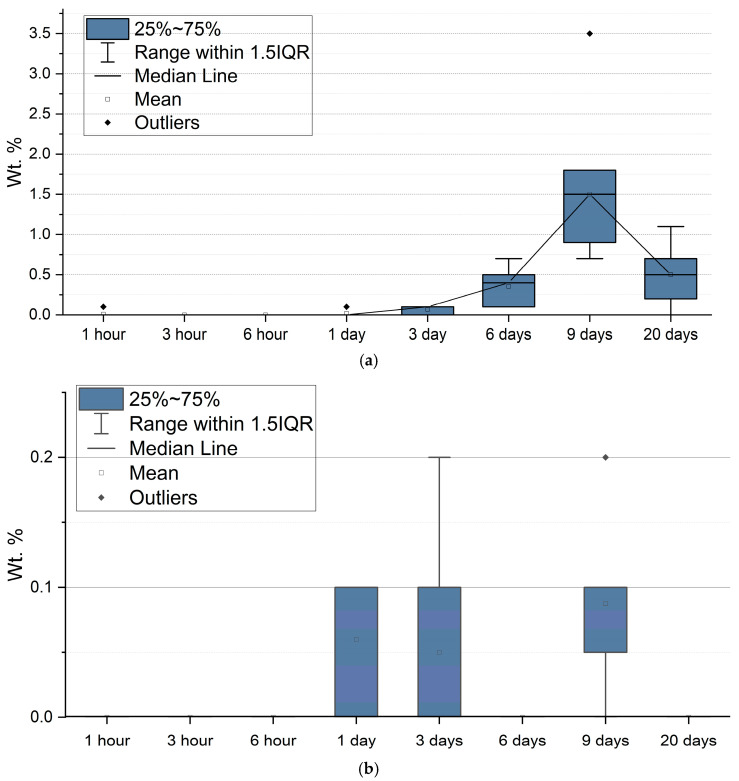
Weight fraction (%) of La on the frustule surfaces of (**a**) *N. shiloi* and (**b**) *H. kolbei* at 50 mg La·L^−1^ in the medium.

## Data Availability

All data used in this study are available upon request from the corresponding author.

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
