# Peer review of "Bioaccumulation of Lanthanum by Two Strains of Marine Diatoms Nanofrustulum shiloi and Halamphora kolbei"

_biology, 2025, doi:10.3390/biology14111489_

Round 1
Reviewer 1 Report
Comments and Suggestions for Authors
This manuscript presents the findings showing the bioaccumulation of lanthanum by two marine diatom strains. This topic is interesting and also instructive for developing marine minimal bioabsorption technologies. This manuscript is well written. Nevertheless, several concerns need to be addressed, as follows:
- Why did the authors choose these two strains and the two lanthanum concentrations (10 mg·L⁻¹ and 50 mg·L⁻¹)? Were there any pre-experiments to support this selection, or is there relevant reference background?
- Methods (Lines 166–176): Before EDX analysis for estimating elemental contents, were the cells pre-treated? What was the concentration of lanthanum remaining in the solution after cell collection? This is also an alternative method for evaluating lanthanum adsorption efficiency.
- Methods (Lines 177–188): Before biomass collection, how were the cultivated cells treated? Were there any washing processes?
- I suggest re-plotting Figures 5, 6, 8, 11, and 12 using Origin rather than Excel (to align with the figure quality of this journal).
- A very important concern is that the authors should clearly distinguish between "adsorption on the cell surface" and "absorption into the cells". Was lanthanum absorbed into the cells? What amounts of lanthanum existed within the cells versus on the cell surface? This distinction (conception) should be considered throughout the manuscript.
Author Response
This manuscript presents the findings showing the bioaccumulation of lanthanum by two marine diatom strains. This topic is interesting and also instructive for developing marine minimal bioabsorption technologies. This manuscript is well written. Nevertheless, several concerns need to be addressed, as follows:
We are very grateful to the Reviewer for his/her attention to our manuscript and for the valuable suggestions. We have carefully addressed each point and made corresponding changes to the text. Below are our point- to-point clarifications.
Comments 1: Why did the authors choose these two strains and the two lanthanum concentrations (10 mg·L⁻¹ and 50 mg·L⁻¹)? Were there any pre-experiments to support this selection, or is there relevant reference background?
Response 1: We thank the reviewer for bringing these points to our attention. We have made additions to the text. We would also like to point out that we have managed to complete the species identification for Halamphora sp. We have replaced the species name in the manuscript with Halamphora kolbei and provided the references used to identify it.
We would like to provide some clarification regarding the choice of concentrations and strains. We chose these two strains of diatoms, Nanofrustulum shiloi and Halamphora kolbei, for our work because they demonstrated stable and quite high growth rates under laboratory conditions. Our goal was to determine whether species influence the bioaccumulation of lanthanum. For this purpose, we selected strains with different lifestyles. N. shiloi is a benthoplanktonic species that is found both in benthos and in plankton, forming colonies with a chained structure. The Halamphora kolbei strain belongs to benthic species and has a seam for attachment to the substrate. In nature, it occurs mainly as single cells, but in culture it forms films attached to the surface of the substrate. In addition, these two strains have different surface areas, as can be seen from the linear dimensions indicated in L. 160, 177, as well as from growth rates, with N. shiloi growing faster than H. kolbei. In addition, it was important for us to include a benthic species in the study, as such species, which have an attached lifestyle, are most valuable as indicators. We have studied the biochemical composition of these strains and proven that they are promising for use in biotechnology. Data on N. shiloi has already been published [1,2], while the manuscript on H. kolbei is being finalised. We expanded our description of the strains in the manuscript and added a justification for their selection in L. 156-181
As a justification for choosing the concentrations used in the study, the following considerations can be mentioned. To study the effect of metals on diatom algae, a metal concentration range of 1 to 100 mg/l is typically used (e.g., for cadmium: [3]). There are no literature data on specific concentrations of rare earth metals affecting diatoms. Therefore, to achieve our goal of identifying the ability of diatom algae to bioaccumulate rare earth metals, we selected a low concentration of La of 10 mg/l in the medium and a high concentration of 50 mg/l from the above range. In addition, using the example of the green microalga Raphidocelis subcapitata, it was shown that at a concentration of lanthanum in the medium of 10 mg/l, its growth slowed down slightly [4]. Further, we chose a concentration of 50 mg/l as potentially toxic to diatoms. Studying the impact of high concentrations of lanthanum is also important for identifying the impact of REEs pollution on the aquatic environment. Although background concentrations of REEs in natural waters are indeed in the nanogram to microgram range, exceptionally high levels of contamination are observed in regions of intensive mining and processing of rare earth elements. For example, in mine waters and tailings pond drainage in REM mining areas (e.g., Baotou, China), total REEs concentrations can reach several milligrams per litre (Romero-Freire et al., 2018; Li et al., 2024). We have added respective explanations to the text in L. 195-202.
Comments 2: Methods (Lines 166–176): Before EDX analysis for estimating elemental contents, were the cells pre-treated? What was the concentration of lanthanum remaining in the solution after cell collection? This is also an alternative method for evaluating lanthanum adsorption efficiency.
Response 2: We thank the Reviewer for the insightful questions. Yes, the pre-treatment was carried out for cells before the EDX analysis. We have added clarification to the text and moved that part to the subsection “2.3. Sample Processing and Microscopy” (L. 252-255).
“For the scanning electron microscopy (SEM, Hitachi SU3500, Tokyo, Japan) analysis and energy dispersive X-ray spectroscopy (EDX), samples were collected with a sterile pipette. One drop from each dish was placed on the cover glass and then rinsed repeatedly with distilled water using an automatic pipette.”
We would like to note that at this stage of research, our goal was only to identify the ability of diatoms to accumulate lanthanum. However, in the future work, we plan to conduct experiments with controlled initial and final metal concentration in the nutrient medium to calculate the total sorption capacity of diatoms. We have incorporated this information in the paragraph discussing research prospects on this matter on L. 982.
Comments 3: Methods (Lines 177–188): Before biomass collection, how were the cultivated cells treated? Were there any washing processes?
Response 3: We thank the reviewer for the questions. Yes, a mandatory step in preparing biomass for the elemental analysis using ICP-MS was washing it to remove any remaining nutrient medium. We added an explanation and moved this information to subsection ‘2.4. Analytical Sample Preparation and Elemental Analysis in Diatom Biomass’ (L.264-269).
Comments 4: I suggest re-plotting Figures 5, 6, 8, 11, and 12 using Origin rather than Excel (to align with the figure quality of this journal).
Response 4: We are very grateful to the reviewer for this recommendation. We have used the OriginPro 2024 SR1 10.1.0.178 software for re-plotting Figures 5, 6, 8, 11, 12 and 13. Information about using this software has been added on L. 247.
Comments 5: A very important concern is that the authors should clearly distinguish between "adsorption on the cell surface" and "absorption into the cells". Was lanthanum absorbed into the cells? What amounts of lanthanum existed within the cells versus on the cell surface? This distinction (conception) should be considered throughout the manuscript.
Response 5: We are grateful to the reviewer for this comment. We have tried to check this aspect and make clarifications in the manuscript (in the Abstract, Materials and Methods, Results and Conclusions sections). In addition, we would like to clarify that the aim of this study was to identify the ability of diatoms to bioaccumulate REE, as there was no direct evidence of this for living diatoms yet. The lanthanum content on the cell surfaces first increased and then decreased, but overall, the content of this element in the biomass was very high. Therefore, it can be assumed that lanthanum accumulated not only on the surfaces of the valves, but also inside the diatom cells, by analogy with other metals. In our further research, we plan to study and compare the accumulation of lanthanum specifically inside the cells.
References
- Blaginina, A.; Ryabushko, L.I. Finding of a rare species of diatom Nanofrustulum shiloi(Lee, Reimer et Mcenery) Round, Hallsteinsen & Paasche, 1999 in the periphyton of the coastal waters of the Black Sea. J. Algae 2021, 23, 247–256. doi:10.1615/InterJAlgae.v23.i3.40.
- Blaginina, A.A.; Zheleznova, S.N.; Miroshnichenko, E.S.; Gevorgiz, R.G.; Ryabushko, L.I. The diatom Nanofrustulum shiloi as a promising species in modern biotechnology. Appl. Biochem. Microbiol. 2024, 60, 483–495. DOI: 10.1134/S0003683824603615
- Torres, E.; Cid, R.; Herrero, C.; Abalde, J. Removal of cadmium ions by the marine diatom Phaeodactylum tricornutum Bohlin accumulation and long-term kinetics of uptake. Bioresource Technology 1998, 63, 213–220.
- Siciliano, A.; Guida, M.; Serafini, S.; Micillo, M.; Galdiero, E.; Carfagna, S.; Salbitani, G.; Tommasi, F.; Lofrano, G.; Suarez, E.G.P.; Gjata, I.; Brouziotis, A.A.; Trifuoggi, M.; Liguori, R.; Race, M.; Fabbricino, M.; Libralato, G. Long-Term Multi-Endpoint Exposure of the Microalga Raphidocelis subcapitata to Lanthanum and Cerium. Sci. Total Environ. 2021, 790, 148229. https://doi.org/10.1016/j.scitotenv.2021.148229
With gratitude,
The Authors

Reviewer 2 Report
Comments and Suggestions for Authors
The manuscript titled Bioaccumulation of lanthanum by two strains of marine diatoms Nanofrustulum shiloi and Halamphora sp is very interesting. However, the following corrections are recommended:
- The abstract of the paper should be made more concise to highlight the key research findings of the research work.
- In the section 2. Materials and Methods the authors have mentioned about the Strains Characteristics and Cultivation Conditions of two strains of diatoms (Bacillariophyta) N. shiloi DIA-N.sh.-KarBay and Halamphora. But the authors have not mentioned the isolation techniques and the identification tools. The information need sot be incorporated with suitable references.
- The discussion should be more elaborated in the light of the recent research findings.
- The conclusion should be made more precise and condensed.
The typological and grammatical errors needs to be corrected.
Author Response
The manuscript titled Bioaccumulation of lanthanum by two strains of marine diatoms Nanofrustulum shiloi and Halamphora sp is very interesting. However, the following corrections are recommended:
We are very grateful to the Reviewer for his/her attention to our manuscript and for the valuable suggestions. We have carefully addressed each point and made corresponding changes to the text. Below are our point- to-point clarifications.
Comments 1: The abstract of the paper should be made more concise to highlight the key research findings of the research work.
Response 1: We have revised the abstract in accordance with the Reviewer's comments. We hope that the text is now more concise and clearer.
Comments 2: In the section 2. Materials and Methods the authors have mentioned about the Strains Characteristics and Cultivation Conditions of two strains of diatoms (Bacillariophyta) N. shiloi DIA-N.sh.-KarBay and Halamphora. But the authors have not mentioned the isolation techniques and the identification tools. The information need sot be incorporated with suitable references.
Response 2: We would like to point out that we have managed to complete the species identification for Halamphora sp. We have replaced the species name in the manuscript with Halamphora kolbei and provided the references used to identify it. We have added information about the isolation techniques and expanded the information about strains in subsection 2.1. Strains Characteristics and Cultivation Conditions.
Comments 3: The discussion should be more elaborated in the light of the recent research findings.
Response 3: We would like to gently bring to the Reviewer's attention that research on the effects of rare earth elements on diatoms is currently limited, and this also applies to the most recent findings. However, taking into account the Reviewer's comment, we rechecked this point and found another recently published work, which we have cited in our manuscript [1]. We have also added more recent sources on the subject of our research [2-4].
Comments 4: The conclusion should be made more precise and condensed.
Response 4: We have completely revised the Conclusions in accordance with the Reviewer's comments.
References
- Sun, S.; Zheng, X.; Liu, H.; Ma, X.; Liu, C.; Bai, J.; Liu, H.; Xia, J.; Wang, J. Toxicity Passivation of Rare Earth Elements via the Diatom-Mediated ‘Mineralisation Self-Regulation’ Mechanism: A Silicon Metabolic Feedback-Driven Pathway for Nanoparticle Formation. J. Environ. Chem. Eng. 2025, 13, 119504. https://doi.org/10.1016/j.jece.2025.119504
- Brzozowska, W.; Wojtczak, I.; Sprynskyy, M. The Use of Diatoms in the Synthesis of New 3D Micro-Nanostructured Composites (SiO2/CaCO3/Corg/NdVO4NPs and SiO2/CaO/Corg/NdVO4NPs Exhibiting an Intense Anti-Stokes Photoluminescence. Materials 2024, 17, 490. https://doi.org/10.3390/ma17020490
- Yanying Zhang, Wenfei Shen, Lili Li, Zaixin Long, Shuxin Li, Tonghui Li, Yao Wang, Olle Inganäs, Jianguo Tang, Living diatoms integrate polysaccharide-Eu3+ complex for UV downconversion, Journal of Materials Research and Technology, Volume 19, 2022, Pages 2774-2780, https://doi.org/10.1016/j.jmrt.2022.05.187;
- Siciliano, A.; Guida, M.; Serafini, S.; Micillo, M.; Galdiero, E.; Carfagna, S.; Salbitani, G.; Tommasi, F.; Lofrano, G.; Suarez, E.G.P.; Gjata, I.; Brouziotis, A.A.; Trifuoggi, M.; Liguori, R.; Race, M.; Fabbricino, M.; Libralato, G. Long-Term Multi-Endpoint Exposure of the Microalga Raphidocelis subcapitata to Lanthanum and Cerium. Sci. Total Environ., 2021, 790, 148229. https://doi.org/10.1016/j.scitotenv.2021.148229
With gratitude,
The Authors

Reviewer 3 Report
Comments and Suggestions for Authors
Overall, this study has some environmental implications. The reviewer has provided the following comments for further improvement of the quality of the manuscript.
Comments:
- Abstract: At the end, add a brief statement on the important implication/contribution of the findings of the present study.
- Keywords: Authors are suggested to avoid listing of keywords which are already present in the manuscript title. Replace them with alternative which are important and relevant to the work as it would maximize findings of their article by the readers in the scientific database.
- Line 54 – 55: “Being “iron-loving” organisms, diatoms have a number…”. Any quantitative data on the iron concentrations in marine water?
- Line 77 – 82: It is better to mention the concentrations of REEs reported in the cited past studies.
- Introduction, in the last paragraph, strengthen the statements on the novelty and importance of this work with respect to the previous literature. Also, state what are the major hypotheses tested in this study.
- Line 126: “The two diatom strains were cultivated in F/2 nutrient medium…”. Provide the detailed components and composition of the medium.
- Line 131 – 132: “at La concentration of 10 mg·l-1 was 7.01 and at 50 mg·l-1…”. Why these specific La concentrations were chosen. Are you mimicking the actual concentration detected in marine environment?
- Before conclusions, add a section describing on the important implications and future perspectives of this work.
- This is very controlled lab-scale experiment. But, in real marine ecosystems, the aqueous conditions would be totally different with salinity, pH and presence of other organic and inorganic ions? Would you expect the similar results as reported in this study? Some discussion needed on this area.
Author Response
Overall, this study has some environmental implications. The reviewer has provided the following comments for further improvement of the quality of the manuscript.
We are very grateful to the Reviewer for his/her attention to our manuscript and for the valuable suggestions. We have carefully addressed each point and made corresponding changes to the text. Below are our point- to-point clarifications.
Comments:
Comments 1: Abstract: At the end, add a brief statement on the important implication/contribution of the findings of the present study.
Response 1: Many thanks to the Reviewer for bringing this to our attention. We have added a sentence to the Abstract describing the potential implication of the revealed ability of diatoms to bioaccumulate REEs on L. 39-42.
Comments 2: Keywords: Authors are suggested to avoid listing of keywords which are already present in the manuscript title. Replace them with alternative which are important and relevant to the work as it would maximize findings of their article by the readers in the scientific database.
Response 2: We thank the Reviewer for this valuable suggestion. We have replaced the keywords in the manuscript.
Comments 3: Line 54 – 55: “Being “iron-loving” organisms, diatoms have a number…”. Any quantitative data on the iron concentrations in marine water?
Response 3: We rephrased this sentence and added a reference to a review discussing iron metabolism strategies in diatoms.
Comments 4: Line 77 – 82: It is better to mention the concentrations of REEs reported in the cited past studies
Response 4: We have provided the ranges of REE concentrations previously identified in the coastal waters of Crimea.
Comments 5: Introduction, in the last paragraph, strengthen the statements on the novelty and importance of this work with respect to the previous literature. Also, state what are the major hypotheses tested in this study.
Response 5: We are thankful to the Reviewer for this valuable suggestion. We have made additions and included our main hypothesis in the last paragraph of the Introduction in L.133-134.
Comments 6: Line 126: “The two diatom strains were cultivated in F/2 nutrient medium…”. Provide the detailed components and composition of the medium.
Response 6: We added the F/2 nutrient composition on L. 185-188.
Comments 7: Line 131 – 132: “at La concentration of 10 mg·l-1 was 7.01 and at 50 mg·l-1…”. Why these specific La concentrations were chosen. Are you mimicking the actual concentration detected in marine environment?
Response 7: We thank the Reviewer for this question. To study the effect of metals on diatom algae, a metal concentration range of 1 to 100 mg/l is typically used (e.g., for cadmium Torres, E.; Cid, R.; Herrero, C.; Abalde, J. Removal of cadmium ions by the marine diatom Phaeodactylum tricornutum Bohlin accumulation and long-term kinetics of uptake. Bioresource Technology 1998, 63, 213–220.). There are no data on specific concentrations of rare earth metals in the literature. Therefore, to achieve our goal of identifying the ability of diatom algae to bioaccumulate rare earth metals, we selected a low La concentration of 10 mg/l in the medium and a high concentration of 50 mg/l from the above range. Using the example of the green microalga Raphidocelis subcapitata, it was shown that at a La concentration of 10 mg/l in the medium, growth slowed down slightly (Siciliano et al., 2021,). Further, we chose a concentration of 50 mg/l as potentially toxic to diatoms. Studying the impact of high concentrations of lanthanum is also important for identifying the impact of REE pollution on the aquatic environment. Although background concentrations of REEs in natural waters are indeed in the range of nanograms to micrograms per litre, exceptionally high levels of contamination are observed in regions of intensive mining and processing of rare earth elements. For example, in mine waters and tailings pond drainage in REE mining areas (e.g., Baotou, China), total REE concentrations can reach several milligrams per litre (Romero-Freire et al., 2018; Li et al., 2024). Anthropogenic pollution of the Rhine River was also detected in the densely populated city of Worms, reaching a concentration of 49 mg/l (Serkan Kulaksız, Michael Bau, 2011). We have added references in L. 108-110 and clarifications to the text in L. 197-202.
Comments 8: Before conclusions, add a section describing on the important implications and future perspectives of this work.
Response 8: We are thankful to the Reviewer for this idea, which significantly improves the manuscript. We have added a paragraph to the manuscript explaining our future intentions to study REE bioaccumulation by diatoms and the prospects for applying this ability in L. 779-788.
Comments 9: This is very controlled lab-scale experiment. But, in real marine ecosystems, the aqueous conditions would be totally different with salinity, pH and presence of other organic and inorganic ions? Would you expect the similar results as reported in this study? Some discussion needed on this area.
Response 9: We thank the Reviewer for this question. We have not found any data in the literature on similar studies imitating natural conditions. There is even a lack of direct evidence of lanthanum bioaccumulation by living diatom cells. Therefore, at this stage, our main goal was to determine whether diatoms are capable of accumulating rare earth elements. Taking into account the Reviewer's comment, we added a brief discussion on this issue in L. 720-731.
With gratitude,
The Authors

Round 2
Reviewer 1 Report
Comments and Suggestions for Authors
I am satisfied with the answers and revisions.
Reviewer 3 Report
Comments and Suggestions for Authors
The submitted revision looks satisfactory. The revised MS can be accepted.